CIAlign: A highly customisable command line tool to clean, interpret and visualise multiple sequence alignments

Tumescheit Charlotte
Firth Andrew E.
Brown Katherine kab84@cam.ac.uk
Department of Pathology, University of Cambridge , Cambridge , United Kingdom
Morgenstern Burkhard
Electronic publication date: 2022 Mar 15
Publication date: 2022
Volume: 10
Electronic Location ID: e12983
Received 2021 Apr 6; Accepted 2022 Feb 1
Copyright: ©2022 Tumescheit et al.
Copyright year: 2022
Copyright holder: Tumescheit et al.
License: This is an open access article distributed under the terms of the Creative Commons Attribution License, which permits unrestricted use, distribution, reproduction and adaptation in any medium and for any purpose provided that it is properly attributed. For attribution, the original author(s), title, publication source (PeerJ) and either DOI or URL of the article must be cited.
License URL: https://creativecommons.org/licenses/by/4.0/

Keywords: Multiple sequence alignment, Alignment quality, Python tool, Comparative genomics, Transcriptomics, Phylogenetics

Funding: Wellcome Trust 106207 European Research Council 646891 This work was supported by the Wellcome Trust (106207) and the European Research Council (646891). The funders had no role in study design, data collection and analysis, decision to publish, or preparation of the manuscript.

==============================
Background

Throughout biology, multiple sequence alignments (MSAs) form the basis of much investigation into biological features and relationships. These alignments are at the heart of many bioinformatics analyses. However, sequences in MSAs are often incomplete or very divergent, which can lead to poor alignment and large gaps. This slows down computation and can impact conclusions without being biologically relevant. Cleaning the alignment by removing common issues such as gaps, divergent sequences, large insertions and deletions and poorly aligned sequence ends can substantially improve analyses. Manual editing of MSAs is very widespread but is time-consuming and difficult to reproduce.

Results

We present a comprehensive, user-friendly MSA trimming tool with multiple visualisation options. Our highly customisable command line tool aims to give intervention power to the user by offering various options, and outputs graphical representations of the alignment before and after processing to give the user a clear overview of what has been removed. The main functionalities of the tool include removing regions of low coverage due to insertions, removing gaps, cropping poorly aligned sequence ends and removing sequences that are too divergent or too short. The thresholds for each function can be specified by the user and parameters can be adjusted to each individual MSA. CIAlign is designed with an emphasis on solving specific and common alignment problems and on providing transparency to the user.

Conclusion

CIAlign effectively removes problematic regions and sequences from MSAs and provides novel visualisation options. This tool can be used to fine-tune alignments for further analysis and processing. The tool is aimed at anyone who wishes to automatically clean up parts of an MSA and those requiring a new, accessible way of visualising large MSAs.

Introduction

Throughout biology, multiple sequence alignments (MSAs) of DNA, RNA or amino acid sequences are often the basis of investigation into biological features and relationships. Applications of MSAs include, but are not limited to, transcriptome analysis, in which transcripts may need to be aligned to genes; RNA structure prediction, in which an MSA improves results significantly compared to predictions based on single sequences; and phylogenetics, where trees are usually created based on MSAs. There are many more applications of MSA at a gene, transcript and genome level, involved in a huge variety of traditional and new approaches to genetics and genomics, many of which could benefit from the tool presented here.

An MSA typically represents three or more DNA, RNA or amino acid sequences, which represent partial or complete gene, transcript, protein or genome sequences. These sequences are aligned by inserting gaps between residues to bring more similar residues (either based on simple sequence similarity or an evolutionary model) into the same column, allowing insertions, deletions and differences in sequence length to be taken into account (Boswell, 1987; Higgins & Sharp, 1988). The first widely used automated method for generating MSAs was Clustal (Higgins & Sharp, 1988) and more recent versions of this tool are still in use today, along with tools such as MUSCLE (Edgar, 2004), MAFFT (Katoh et al., 2002), T-coffee (Notredame, Higgins & Heringa, 2000) and many more. The majority of tools are based upon various heuristics used to optimise progressive sequence alignment using a dynamic programming based algorithm such as the Needleman-Wunsch algorithm (Needleman & Wunsch, 1970).

It has been shown previously that removing divergent regions from an MSA can improve the resulting phylogenetic tree (Talavera & Castresana, 2007). Various tools are available to identify or remove poorly aligned columns, including trimAl (Capella-Gutiérrez, Silla-Martínez & Gabaldón, 2009), Gblocks (Talavera & Castresana, 2007) and ZORRO (Wu, Chatterji & Eisen, 2012). These four tools use various algorithms to assign confidence scores for each column in an MSA. Gblocks (Talavera & Castresana, 2007) identifies and removes stretches of contiguous columns with low conservation. All positions with gaps, or adjacent to gaps, are also removed (Talavera & Castresana, 2007). With trimAl (Capella-Gutiérrez, Silla-Martínez & Gabaldón, 2009), poorly aligned columns are identified using proportion of gaps, residue similarity and consistency across multiple alignments, either column-by-column or based on a sliding window across the alignment. ZORRO uses hidden Markov models to model sequence evolution and calculates posterior probabilities that columns are correctly aligned (Wu, Chatterji & Eisen, 2012). All of these tools have been shown to improve the accuracy of phylogenetic analysis under some circumstances and all can be valuable (Talavera & Castresana, 2007; Capella-Gutiérrez, Silla-Martínez & Gabaldón, 2009; Wu, Chatterji & Eisen, 2012). However, poorly aligned columns are not the only issue found in MSAs. All of these tools are designed to identify problematic columns, but none are able to identify problematic rows which are disrupting an alignment. They also cannot distinguish which gaps are the result of insertions within sequences and which are the result of partial sequences. Column-wise tools can also be too stringent when working with highly divergent alignments. Gblocks, trimAl and ZORRO are specifically tailored towards phylogenetic analysis rather than other applications such as building consensus sequences, scaffolding of contigs or secondary structure analysis.

Various refinement methods incorporated into alignment software can also improve MSAs (Edgar, 2004; Katoh et al., 2002). Some tree building software can also take into account certain discrepancies in the alignment, for example RaXML  (Stamatakis, 2014) can account for missing data in some columns and check for duplicate sequence names and gap-only columns; similarly GUI based toolkits for molecular biology such as MEGA (Kumar et al., 2018) sometimes have options to delete or ignore columns containing gaps.

Several common issues affect the speed, complexity and reliability of specific downstream analyses but are not addressed by these existing tools. Clean and Interpret Alignments (CIAlign) is primarily intended to address four such issues and to be used (where appropriate) in combination with existing tools which remove unreliable alignment columns. Researchers in many fields regularly edit MSAs by hand to address these issues, however, as well as being extremely time consuming, ensuring reproducibility with this approach is almost impossible and it cannot be incorporated into an automated analysis pipeline. CIAlign automatically removes full columns and full or partial rows from user generated MSAs to address these issues in a fast, reproducible manner and can be easily added to an automated pipeline. The downstream applications of alignments cleaned with CIAlign are not limited to phylogenetic analysis and are too numerous to list, but CIAlign as an alignment cleaning tool is particularly targetted towards users working with complex or highly divergent alignments, partial sequences and problematic assemblies and towards those developing complex pipelines requiring fine-tuning of parameters to meet specific criteria.

The first issue we intend to address is that it is common for an MSA to contain more gaps towards either end than in the body of the alignment. This problem occurs at both the sequencing and alignment stage. For example, the ends of de novo assembled transcripts tend to have lower read coverage (Bushmanova et al., 2019) and so have a higher probability of mis-assembly and therefore mis-alignment. MSAs created using these sequences therefore also have regions of lower reliability towards either end. Similarly, both Sanger sequences and sequences generated with Oxford Nanopore’s long read sequencing technology, which are often used directly in MSAs, tend to have lower quality scores at either the beginning or the end  (Richterich, 1998; Tyler et al., 2018; Magi, Giusti & Tattini, 2017). Automated removal of these regions from MSAs would therefore increase the reliability of downstream analyses. As sequences are often partial, poor quality sequence ends can be scattered throughout the alignment, and so do not necessarily result in whole columns which are unreliable. A tool such as CIAlign, which identifies gaps at the ends of sequences on a row-by-row basis, is therefore needed in these cases, rather than a tool which works on whole columns only. Also, while generating an MSA, terminal gaps complicate analysis, and the weighting of terminal gaps relative to internal gap opening and gap extension penalties can make a large difference to the resulting alignment (Fitch & Smith, 1983). This again leads to regions of ambiguity and therefore gaps towards the ends of sequences within the alignment, which can be rectified with CIAlign.

Secondly, insertions or other stretches of sequence can be present in a minority of sequences in an MSA, leading to large gaps in the remaining sequences. For example, alignments of sections of bacterial genomes often result in long gaps representing genes which are absent in the majority of species. These gaps can be observed, for example, in multiple genome alignments shown in Tettelin et al. (2005) for Streptococcus agalactiae and Hu et al. (2011) for Burkholderia, amongst others, which show many genes which are present in only a few genomes. While these regions are of interest in themselves and certainly should not be excluded from all further analysis, they are not relevant for every downstream analysis. For example, a consensus sequence for these bacteria would exclude these regions and their presence would increase the time required for phylogenetic analysis without necessarily adding any additional information. Large gaps in some sequences may also result from missing data, rather than true biological differences and, if this is known to be the case, it is often appropriate to remove these regions before performing phylogenetic analysis (Sayyari, Whitfield & Mirarab, 2017). Unlike other available tools, CIAlign can distinguish between gaps within the body of a sequence, which users may wish to remove, and gaps padding the ends of sequences of different lengths, which occur, for example, when aligning overlapping partial sequences, and remove the internal insertions only.

Thirdly, one or a few highly divergent sequences can heavily disrupt the alignment and therefore complicate downstream analysis. It is very common for an MSA to include one or a few outlier sequences which do not align well with the majority of the alignment. One example of this is in analyses identifying novel sequences in large numbers of datasets. It is common to manually remove phylogenetic outliers which are unlikely to truly represent members of a group of interest (see for example Schulz et al., 2020; Käfer et al., 2019; Bäckström et al., 2019) but this is not feasible when processing large numbers of alignments. Alignment masking tools such as trimAl and Gblocks work column-by-column, and so, unlike CIAlign, are not able to remove divergent rows.

Finally, very short partially overlapping sequences cannot always be reliably aligned using standard global alignment algorithms. It is very common to remove these sequences, manually or otherwise, prior to further analysis.

There are also several common issues in alignment visualisation. Large alignments can be difficult to visualise and a small and concise but accurate visualisation can be useful when presenting results, so this has been incorporated into the software. With many alignment trimming tools it can be difficult to track exactly which changes the software has made, so a visual output showing these changes could be helpful.

Transparency is often an issue with bioinformatics software, with poor reporting of exactly how a file has been processed (Petyuk, Gatto & Payne, 2019; Brito et al., 2020; Langille, Ravel & Fricke, 2018). CIAlign has been developed to process alignments in a transparent manner, to allow the user to clearly and reproducibly report their methodology.

CIAlign is freely available at http://github.com/KatyBrown/CIAlign.

Materials & Methods

CIAlign is a command line tool implemented in Python 3. It can be installed either via pip3 or from GitHub and is independent of the operating system. It has been designed to enable the user to remove specific issues from an MSA, to visualise the MSA (including a markup file showing which regions and sequences have been removed) and to interpret the MSA in several ways. CIAlign works on nucleotide or amino acids alignments and will detect which of these is provided. A log file is generated to show exactly which sequences and positions have been removed from the alignment and why they were removed. Users can then adjust the software parameters according to their needs.

CIAlign takes as its input any pre-computed alignment in FASTA format containing at least two sequences (for some cleaning functions three sequences are required). Most MSAs created with standard alignment software will be of an appropriate scale, for example single or multi-gene alignments and whole genome alignments for many microbial species.

The path to the alignment file is the only mandatory parameter. Every function is run only if specified in the parameters and many function-specific parameters allow options to be fine-tuned. Using the parameter option all will turn on all the available functions and run them with the default parameters, unless otherwise specified. The clean option will run all cleaning functions, visualise all the visualisation functions and interpret all the interpretation functions, again with the default parameters. Additionally, the user can provide parameters via a configuration file instead of via the command line.

CIAlign has been designed to maximise usability, reproducibility and reliability. The code is written to be as readable as possible and all functions are fully documented. All functions are covered by unit tests. CIAlign is freely available, open source and fully version controlled.

Cleaning alignments

CIAlign consists of several functions to clean an MSA by removing commonly encountered alignment issues. All of these functions are optional and can be fine-tuned using user parameters. All parameters have default values. The available functions are presented here in the order they are executed by the program. The order can have a direct impact on the results, the functions removing positions that lead to the greatest disruptions in the MSA should be run first, as they potentially make removing more positions unnecessary and therefore keep processing to a minimum. For example, divergent sequences often contain many insertions compared to the consensus, so removing these sequences first reduces the number of insertions which need to be removed. Sequences can be made shorter during processing with CIAlign and therefore too short sequences are removed last.

Figure 1 shows a graphical representation of an example toy alignment before (Fig. 1A) and after (Figs. 1B–1F) using each function individually. The remove gap only function is run by default after every cleaning step, unless otherwise specified by the user.

Figure 1 Mini alignments showing the main functionalities of CIAlign based on Example 1.

(A) Input alignment before application of CIAlign, generated using the command “CIAlign infile example1.fasta plot_input”. (B) Output alignment showing the functionality of the remove divergent function, generated using the command “CIAlign infile example1.fasta remove_divergent plot_output”. (C) Output alignment showing the functionality of the remove insertions function, generated using the command “CIAlign infile example1.fasta remove_insertions plot_output”. (D) Output alignment showing the functionality of the crop ends function, generated using the command “CIAlign infile example1.fasta crop_ends plot_output”. (E) Output alignment showing the functionality of the remove short sequences function, generated using the command ”CIAlign infile example1.fasta remove_short plot_output”. (F) Output alignment showing the functionality of the remove gap only function, generated using the command “CIAlign infile example1.fasta plot_output”. Subplots were generated using the draw mini alignment function of CIAlign. In all subplots sequences are labelled according to their position in the input alignment.

Remove divergent

For each column in the alignment, this function finds the most common nucleotide or amino acid and generates a temporary consensus sequence. Each sequence is then compared individually to this consensus sequence. Sequences which match the consensus at a proportion of positions less than a user-defined threshold (default: 0.65) are excluded from the alignment (Fig. 1B). It is recommended to run the make similarity matrix function to calculate pairwise similarity before removing divergent sequences, in order to adjust the parameter value for more or less divergent alignments. This function requires an alignment of three or more sequences.

Remove insertions

In order for CIAlign to define a region as an insertion, an alignment gap must be present in the majority of sequences and flanked by a minimum number of non-gap positions on either side, which can be defined by the user (default: 5). This pattern can be the result of an insertion in a minority of sequences or a deletion in a majority of sequences. The minimum and maximum size of insertion to be removed can also be defined by the user (default: 3 and 200, respectively) (Fig. 1C). This function requires an alignment of three or more sequences.

Crop ends

The crop ends function redefines where each sequence starts and ends, based on the ratio of the numbers of gap and non-gap positions observed up to a given position in the sequence. It then replaces all non-gap positions before and after the redefined start and end, respectively, with gaps. This will be described for redefining the sequence start, however crop ends is also applied to the reverse of the sequence to redefine the sequence end. The number of gap positions separating every two consecutive non-gap positions is compared to a threshold and if that difference is higher than the threshold, the start of the sequence will be reset to that position. This threshold is defined as a proportion of the total sequence length, excluding gaps, and can be defined by the user (default: 0.05) (Figs. 1D, 2). The user can set a parameter that defines the maximum proportion of the sequence for which to consider the change in gap positions (default: 0.1) and therefore the innermost position at which the start or end of the sequence may be redefined. It is recommended to set this parameter no higher than 0.1, since even if there are a large number of gap positions beyond this point, this is unlikely to be the result of incomplete sequences (Fig. 2). This function requires an alignment of three or more sequences.

Figure 2 Crop ends diagram.

This manually created example illustrates how the crop ends function works internally. The length of the sequence shown is 111 including gaps and 80 excluding gaps (1). With a threshold of 10% for the proportion of non-gap positions to consider for change in end positions, 8 positions at the start and at the end, respectively, are being considered (illustrated by red crossbars). For each of these, the number of preceding gaps is calculated (2). Then the change in gap numbers (3) for every two consecutive non-gap positions is compared to the gap number change threshold, which is 5%, i.e. 4 gaps, as a default value. Looking at the change in gap numbers, the last change at each end equal to or bigger than the threshold is coloured in red. This leads to redefining the start and the end of this example sequence to be where the nucleotides are coloured in green.

Remove short sequences

The remove short function removes sequences which have less than a specified number of non-gap positions,which can be set by the user (default: 50) (Fig. 1E).

Remove gap only columns

The remove gap only function only removes columns that contain only gaps. These could be introduced by manual editing of the MSA before using CIAlign or by running the functions above (Fig. 1F). The main purpose of the function is to clean the gap only columns that are likely to be introduced after running any of the cleaning functions.

Visualisation

There are several ways of visualising the alignment, which both allow the user to interpret the alignment and clearly show which positions and sequences CIAlign has removed. CIAlign can also be used simply to visualise an alignment, without running any of the cleaning functions. All visualisations can be output as publication ready image files.

Mini alignments

CIAlign provides functionality to generate mini alignments, in which an MSA is visualised using coloured rectangles on a single x and y axis, with each rectangle representing a single nucleotide or amino acid (e.g., Fig. 1, 3–5). Even for large alignments, this function provides a visualisation that can be easily viewed and interpreted. Many properties of the resulting file (dimensions, DPI, file type) are parameterised. In order to minimise the memory and time required to generate the mini alignments, the matplotlib imshow function (Hunter, 2007) for displaying images is used. Briefly, each position in each sequence in the alignment forms a single pixel in an image object and a custom dictionary is used to assign colours. The image object is then stretched to fit the axes.

Figure 3 Mini alignments and legends showing further functionalities of CIAlign based on Example 1.

(A) Alignment showing the functionality of the plot markup function, generated using the command “CIAlign infile example1.fasta all”. The areas that have been removed are marked up in different colours, each corresponding to a certain function of CIAlign. (B) Output alignment after application of all functions of CIAlign combined, generated using the command “CIAlign infile example1.fasta all”. Subplots were generated using the draw mini alignment function.

Figure 4 Mini alignments showing the main functionalities of CIAlign based on Example 2.

(A) Input alignment before application of CIAlign, generated using the command “CIAlign infile example2.fasta plot_input”. (B) Alignment markup showing areas that were removed by CIAlign, generated using the command “CIAlign infile example2.fasta all”. (C) Output alignment after application of CIAlign, generated using the command “CIAlign infile example2.fasta all”. Subplots were generated using the draw mini alignment function.

Figure 5 Mini alignments showing the main functionalities of CIAlign based on Example 3.

(A) Input alignment before application of CIAlign, generated using the command “CIAlign infile example3.fasta plot_input”. (B) Output alignment after application of CIAlign, generated using the command “CIAlign infile example3.fasta all remove_divergent_minperc 0.5”. Subplots were generated using the draw mini alignment function.

Sequence logos

CIAlign can generate traditional sequence logos (Schneider & Stephens, 1990) or sequence logos using rectangles instead of letters to show the information and base/amino acid content at each position, which can increase readability in less conserved regions. Sequence logos can also be generated for sections of the alignment if a set of boundary coordinates is provided.

Interpretation

Some additional functions are provided to further interpret the alignment, for example plotting the number of sequences with non-gap residues at each position (the coverage), calculating a pairwise similarity matrix and generating a consensus sequence with various options.

Given the toy example shown in Fig. 1A, running all possible cleaning functions will lead to the markup plot shown in Fig. 3A and the result shown in Fig. 3B. In the markup plot each removed part is highlighted in a different colour corresponding to the function with which it was removed.

Example alignments

Four example alignments are provided within the software directory to demonstrate the functionality of CIAlign. Examples 1 and 2 use simulated sequences, Examples 3 and 4 use real biological sequences and are designed to resemble the type of complex alignment many researchers encounter.

Example 1 is a very short alignment of six sequences which was generated manually by creating arbitrary sequences of nucleotides that would show every cleaning function while being as short as possible. This alignment contains an insertion, gaps at the ends of sequences, a very short sequence and some highly divergent sequences.

Example 2 is a larger alignment based on randomly generated amino acid sequences using RandSeq (a tool from ExPASy (Gasteiger et al., 2003)) with an average amino acid composition, which were aligned with MAFFT v7.407, under the default settings (Katoh et al., 2002). The sequences were adjusted manually to reflect an alignment that would fully demonstrate the functionalities of CIAlign. It consists of many sequences that align well, however there are again a few problems: one sequence has a large insertion, one is very short, one is extremely divergent and some have multiple gaps at the start and at the end.

For Example 3, putative mitochondrial gene cytochrome C oxidase I (COI) sequences were identified by applying TBLASTN v2.9.0 (Camacho et al., 2009) to the human COI sequence (GenBank accession NC_012920.1, positions 5,904–7,445, translated to amino acids), querying against 1,565 transcriptomic datasets from the NCBI transcriptome shotgun assembly (TSA) database (Transcriptome Shotgun Assembly Sequence Database, https://www.ncbi.nlm.nih.gov/genbank/tsa/) under the default settings. 2,855 putative COI transcripts were reverse complemented where required, and those corresponding to the COI gene of the primary host of the TSA dataset were identified using the BOLD online specimen identification engine  (Ratnasingham & Herbert, 2007) (accessed 07/10/2019) querying against the species level barcode records. The resulting 232 sequences were then aligned with MAFFT, under the default settings.

For Example 4, 91 sequences were selected from Example 3 to be representative of as many taxonomic families as possible and to exclude families with unclear phylogeny in the literature. These sequences were aligned with MAFFT under the default settings and the alignment was refined with 1000 iterations. Robinson-Foulds (RF) distances (Robinson & Foulds, 1981) of the resulting trees were calculated using ete3 compare (Huerta-Cepas, Serra & Bork, 2016).

Materials and methods for benchmarking and for larger scale examples with biological data are provided as Materials and Methods S1.

Results

Here, an example is presented and the visualisation functions are used to illustrate the functionality of CIAlign. Results will differ when using different parameters and thresholds.

CIAlign was applied to the Example 2 alignment with the following options:

python3 CIAlign.py infile INFILE outfile_stem OUTFILE_STEM all

Using these settings on the alignment in Fig. 4A results in the markup shown in Fig. 4B and the output shown in Fig. 4C. The markup shows which function has removed each sequence or position. The benefits of CIAlign are clear in this simulation—the single poorly aligned sequence, the large insertion, very short sequences and gap-only columns have been removed and the unreliably aligned end segments of the sequences have been cropped. The resulting alignment is significantly shorter, which will speed up and simplify any further analysis. The clear graphical representation makes it easy to see what has been removed, so in the case of over-trimming the user can intervene and adjust functions and parameters.

In order to demonstrate the use of CIAlign on real biological sequences, an alignment was generated based on the COI gene commonly used in phylogenetic analysis and DNA barcoding (Ratnasingham & Herbert, 2007). As CIAlign addresses some common problems encountered when generating an MSA based on de novo assembled transcripts, which tend to have a higher error rate at transcript ends, gaps due to difficult to assemble regions and divergent sequences due to chimeric connections between unrelated regions (Bushmanova et al., 2019; Liao et al., 2019), COI-like transcripts were identified by searching the NCBI transcriptome shotgun assembly database. Aligning these transcripts demonstrated several common problems—multiple insertions, poor alignment at the starts and ends of sequences and a few divergent sequences resulting in excessive gaps (Fig. 5A). This alignment was cleaned using the default CIAlign settings except the threshold for removing divergent sequences was reset to 50%, as some of the sequences are from evolutionarily distant species. Cleaning this alignment with CIAlign took an average of 68.1 s and used on average a maximum of 1.13GB of RAM (mean across 10 runs, on one Intel Core i7-7560U core with 4 GB of RAM, running at 2.40 GHz, RAM measured as maximum resident set size, this machine and 10 replicates were also used for all subsequent measurements of CIAlign resource requirements in this section). Under these settings, CIAlign resolved several of the problems with the alignment: the insertions and highly divergent sequences were removed and the poorly aligned regions at the starts and ends of sequences were cropped (Fig. 5B). One sequence and 6,029 positions were removed from the alignment and a total of 2,446 positions were cropped from the ends of 112 sequences. The processed alignment is 26.6% of the size of the input alignment. However, a minimal amount of actual sequence data (as opposed to gaps) was removed, with 85.7% of bases remaining.

A subset of this sequence set was selected to demonstrate the functionality of CIAlign in streamlining phylogenetic analysis. 91 COI-like transcripts from different taxonomic families of metazoa were selected from Example 3, incorporated into an MSA and cleaned using CIAlign with the same settings as above (Fig. S1). CIAlign took an average of 20.8 s to clean this alignment and used, on average, a maximum of 486 MB of RAM. 1,437 positions were removed from the alignment and a total of 289 positions were cropped from the ends of 17 sequences. The processed alignment is 70.7% of the size of the input alignment and 96.5% of bases remain. Phylogenetic trees were generated for the input alignment and for the alignment processed with CIAlign, using PhyML (Guindon & Gascuel, 2003) under the GTR model plus the default settings. For the input alignment, PhyML used 138 MB of memory and took 532 s . For the cleaned alignment PhyML used 109 MB of memory and took 243 s. The tree generated with the input alignment (Fig. S1D) had an RF distance from a “correct” tree (generated manually based on the literature, Fig. S1D, literature listed in Materials and Methods S1) of 100 (normalised Robinson-Foulds (n-RF) 0.570, (QD) (Smith, 2019) 0.159). The tree generated with the cleaned alignment (Fig. S1E) had an RF distance from the correct tree of 90 (n-RF 0.520, QD 0.073) Therefore, the tree based on the CIAlign cleaned alignment was generated more quickly and was more similar to the expected tree.

Testing with simulated and benchmark data

EvolvAGene, INDELible and BAliBASE—alignment and phylogeny

We performed a series of benchmarking analyses on simulated and benchmark data, in order to test and demonstrate the utility of the CIAlign cleaning functions, confirm the validity of our default parameter settings and ensure that running these functions does not have unexpected negative effects on downstream analyses. Running any tool which removes residues from an alignment has a potential cost, so these tests are intended to allow users to weigh this against the benefit of running CIAlign for their intended use case.

First, CIAlign was tested using three tools: EvolvAGene (Hall, 2008), INDELible  (Fletcher & Yang, 2009) and BAliBASE (Bahr et al., 2001). EvolvAGene and INDELible generate sets of unaligned sequences alongside “true” alignments and phylogenies expected to accurately represent the relationship between the sequences  (Hall, 2008; Fletcher & Yang, 2009). BAliBASE is a set of alignments designed for benchmarking sequence alignment tools (Bahr et al., 2001). We used these tools to determine if cleaning a user generated alignment with CIAlign affects its distance from the true alignment.

Figure 6 Metrics from benchmarking CIAlign with simulated data.

(A) Box plots showing the impact of running CIAlign cleaning functions with relaxed (green, R, left box), moderate (blue, M, middle box) and stringent (pink, S, right box) parameter values on alignments of sequences simulated using either EvolvAGene (Bahr et al., 2001) or INDELible (Sievers & Higgins, 2018) and on the BAliBASE (Thompson, Plewniak & Poch, 1999) benchmark alignments (plots are combined for the three tools, for separated plots see Fig. S3). From left to right, the y-axis represents proportion of correctly aligned pairs of residues (Sievers et al., 2013) removed (identified by comparison with a benchmark alignment), proportion of total nucleotides (i.e. non-gap positions) removed, proportion of gaps removed, proportion of positions (gap or non-gap) removed. (B) Scatter plot showing a linear regression analysis of the impact of the total proportion of positions removed on the proportion of correctly aligned pairs of residues removed by CIAlign for relaxed, moderate and stringent parameter values. The statistic m is the slope of the regression line. (C) Violin plots showing the distribution of normalised Robinson-Foulds distances (Hall, 2008) (left column) and Quartet divergence (right column) (Fletcher & Yang, 2009) between benchmark trees and test trees without running CIAlign cleaning functions (yellow) and after running CIAlign with the three sets of parameter values, for trees based on simulated sequences generated with EvolvAGene (Bahr et al., 2001) (top row) and INDELible (Sievers & Higgins, 2018) (bottom row). Red and black lines show the median and mean, respectively. (D) Density plot showing the distribution of the percentage identity between the input sequence to EvolvAGene (Bahr et al., 2001) and a consensus sequence based on an alignment of the simulated sequences generated by this tool, without running CIAlign (yellow) and after running CIAlign cleaning functions with the three sets of parameter values. (E) Density plots showing the distribution of the percentage identity between the input sequence to BadRead (Sievers & Higgins, 2020) and a consensus sequences generated with (blue) and without (yellow) running CIAlign cleaning functions for alignments of good (top), medium (middle) and poor (bottom) quality simulated reads. (F) Box plot showing the proportion of correct positions removed by the CIAlign cleaning functions for alignments of good, medium and bad quality simulated reads (left) and scatter plot showing a linear regression analysis of the impact of the total proportion of positions removed on the proportion of correct residues removed by CIAlign for each read quality level (right). The statistic m is the slope of the regression line. (G) Box plots showing the impact of running CIAlign on the mean ZORRO (Wu, Chatterji & Eisen, 2012) column confidence score (top) and the proportion of columns with high ZORRO column confidence scores (>0.4) for EvolvAGene (Bahr et al., 2001) (left), INDELible (Sievers & Higgins, 2018) (centre) and BAliBASE (Thompson, Plewniak & Poch, 1999) (right) alignments.

Test alignments were created using four common alignment algorithms—Clustal Omega (Sievers & Higgins, 2018), MUSCLE (Edgar, 2004), MAFFT global (FFT-NS-i)  (Katoh et al., 2002) and MAFFT local (L-NS-i) (Katoh et al., 2002). These alignments were then cleaned with CIAlign with relaxed, moderate or stringent parameter settings (Table S1). With relaxed CIAlign settings, a median of 0.400% of correct pairs of aligned residues (POARs) (Thompson, Plewniak & Poch, 1999) were removed, for moderate settings 2.31% were removed and for stringent settings 6.06% were removed (Fig. 6A, Table 1). For comparison, the median total proportion of residues removed was 2.38% for relaxed, 3.24% for moderate and 5.36% for stringent (Fig. 6A, Table 1). The median proportions of gap positions removed were much higher: 51–56% for all sets of parameters (Fig. 6A, Fig. S2, Table 1). This shows that with relaxed and moderate settings, running CIAlign has a very minimal impact on correctly aligned residues in the alignment, while a considerable amount of gaps and noise are removed. The more stringent settings should be used cautiously, however, even with high stringency, a large majority of correctly aligned residues remain and the majority of gaps are removed. These results are separated by simulation tool (EvolvAGene, INDELible or BAliBASE) and alignment tool (MUSCLE, MAFFT global, MAFFT local and Clustal Omega) in Fig. S2.

To directly compare the impact of CIAlign on correctly aligned pairs of residues to its overall impact, we fitted a linear regression line to show how, on average, the overall proportion of positions removed from the alignment impacts the proportion of correctly aligned residues reoved (Fig. 6B). The resulting line had a gradient of 0.281 for relaxed parameters, 0.361 for moderate parameters and 0.554 for stringent parameters. In other words, for every 1% of material removed from the alignment by CIAlign with relaxed settings, an average of only 0.281% of correctly aligned residue pairs will be removed, with moderate settings 0.361% and with stringent settings 0.554% (Fig. 6B). This will vary depending on the input alignment and the use case. These results are shown separately for MUSCLE, MAFFT and Clustal Omega in Fig. S2E. The impact of CIAlign on correctly aligned pairs is most severe on the Clustal Omega EvolvAGene alignments, which have lower pairwise identity than the alignments generated with the other tools and so have more sequences removed entirely by the remove divergent function (discussed below).

Table 1 Table showing the impact of running CIAlign cleaning functions with relaxed, moderate and stringent parameter values on alignments of sequences simulated using either EvolvAGene  (Bahr et al., 2001) or INDELible (Sievers & Higgins, 2018) and on the BAliBASE (Thompson, Plewniak & Poch, 1999) benchmark alignments (results are combined for the three tools).

For each stringency level, the median percentage of correctly aligned pairs of residues (Sievers et al., 2013) removed (identified by comparison with a benchmark alignment), proportion of total nucleotides (i.e., non-gap positions) removed, proportion of gaps removed and proportion of positions (gap or non-gap) removed have been calculated for EvolvAGene, INDELible and BAliBASE. The mean normalised Robinson-Foulds (RF) distance (Hall, 2008) and Quartet divergence (Fletcher & Yang, 2009) are based on comparison with benchmark trees for EvolvAGene and INDELible. Consensus percentage identity is between the input sequence to EvolvAGene and a consensus sequence based on an alignment of the simulated sequences generated by this tool. Confidence scores are the mean ZORRO (Wu, Chatterji & Eisen, 2012) column confidence scores and the proportion of columns with high ZORRO column confidence scores (>0.4) for EvolvAGene, INDELible (Sievers & Higgins, 2018) and BAliBASE (Thompson, Plewniak & Poch, 1999) alignments. All statistics are two-sided Mann Whitney U tests comparing the alignment without running CIAlign to the alignment after running CIAlign with the specified parameters.

Metric	Statistic	CIAlign stringency	
		None	Relaxed	Moderate	Stringent	
Correct Pairs Removed	Median %	–	0.400	2.31	6.06	
Nucleotides Removed	Median %	–	2.38	3.24	5.36	
Gaps Removed	Median %	–	51.7	52.0	55.9	
Positions Removed	Median %	–	9.62	10.6	13.3	
Normalised RF Distance	Mean	0.241	0.240	0.246	0.250	
MWU Test Statistic	–	320490	316553	312115	
MWU P-value	–	0.955	0.695	0.394	
Significance	–	–	–	–	
Quartet Divergence	Mean	0.162	0.163	0.167	0.171	
MWU Test Statistic	–	320125	316179	311455	
MWU P-value	–	0.989	0.665	0.356	
Significance	–	–	–	–	
Consensus Percentage Identity	Mean	67.2	71.5	71.5	71.5	
MWU Test Statistic	–	23294	22924	23258	
MWU P-value	–	1.89E -67	2.61E -68	1.56E -67	
Significance	–	***	***	***	
Confidence Score	Mean	3.66	4.68	4.63	4.72	
MWU Test Statistic	–	688583	700927	688059	
MWU P-value	–	8.65E -31	7.84E -28	3.61E -33	
Significance	–	***	***	***	
Percentage High Confidence Columns	Mean	69.1	84.3	84.0	85.6	
MWU Test Statistic	–	465471	477660	462908	
MWU P-value	–	2.44E -111	1.31E -105	6.89E -116	
Significance	–	***	***	***	
Notes.

Significance is shown as *** if the p-value is less than 0.001, ** if the p-value is less than 0.01, * if the p-value is less than 0.05 and–if the p-value is greater than 0.05.

In most cases, CIAlign is not intended or expected to change the phylogenetic tree resulting from an alignment, although in many cases it will make building phylogenetic trees faster. To test this, phylogenetic trees were generated for each of the EvolvAGene and INDELible alignments (BAliBASE does not provide reference trees) to determine if cleaning with CIAlign impacts the distance between the true phylogenetic tree and a phylogenetic tree based on a test alignment (Fig. 6C, Table 1). For the EvolvAGene and INDELible alignments, the mean n-RF distance and QD between the test trees and true trees were virtually unchanged by running CIAlign and none of the changes were statistically significant (n-RF p = 0.955, 0.695, 0.394, QD p = 0.989, 0.665, 0.356 for relaxed, moderate, stringent, respectively, Mann Whitney U test) (Fig. 6C, Table 1).

We also compared the input sequence for our EvolvAGene simulations to consensus sequences based on alignments with and without CIAlign cleaning. For all three stringency levels, CIAlign increased the percentage nucleotide identity between the consensus sequence and the input sequence by between 4% and 5% (Fig. 6D, Table 1). All of these changes are statistically significant (relaxed: p = 1.89E−67, moderate: p = 2.61E−68, stringent, p = 1.56E−67, Mann–Whitney U test).

The long-read sequencing simulation tool BadRead (Wick, 2019) was used to demonstrate the use of CIAlign to remove common sources of error in long read sequencing data. Sequences were generated to represent low, moderate and high quality Oxford Nanopore reads based on an input genome, then aligned and cleaned with CIAlign with moderate settings (Table S1). Using CIAlign increased the identity between the alignment consensus and the input sequence significantly for all read quality levels—by 6.57% for high quality reads, 9.51% for moderate quality reads and 12.3% for poor quality reads (Fig. 6E, Table S2) (p = 2.22E−35, 1.37E−13, 1.55E−9, respectively, Mann–Whitney U test). For the high quality reads, the reads cleaned with CIAlign generated consensus sequences almost identical to the input sequence, with a mean of 99.2% identity (Fig. 6E, Table S2). The proportion of the positions removed from the alignment which were correct (in this case positions in the alignment which match the input sequence used to generate the reads) was calculated in order to demonstrate the potential cost of running CIAlign. For the good quality simulated reads, a median of 3.99% of the positions which were removed match the input sequence, for medium quality 5.03% and for low quality 7.31% (Fig. 6F, Table S2). A linear regression analysis showed that, on average, removing 1% of total positions with CIAlign removes 0.0740% of correct positions for good quality simulated reads, 0.504% for medium quality reads and 0.491% for bad quality reads (Fig. 6F).

The alignment masking tool ZORRO (Wu, Chatterji & Eisen, 2012) provides a confidence score (maximum 10) for each column in the MSA, representing a measure of uncertainty in that column. This confidence score was measured for each column of each of the EvolvAGene, INDELible and BAliBASE alignments. The mean confidence score increased by 1.02 for relaxed, 0.970 for moderate and 1.06 for stringent CIAlign settings, all of which are significant improvements (p = 8.65E−31, 7.84e−28, 3.61E−33, respectively, Mann–Whitney U test) (Fig. 6G). The proportion of columns with a confidence score greater than 0.4 (the minimum suggested in the ZORRO documention (Wu, Chatterji & Eisen, 2012)) was also measured and increased by 15.2%, 14.9% and 16.5% for relaxed, moderate and stringent CIAlign settings (p = 2.44E−111, 1.31E−105, 6.88E−116, respectively, Mann–Whitney U test) (Fig. 6G, Table 1).

HomFam—alignment and phylogeny

CIAlign was also benchmarked using the HomFam (Sievers et al., 2013) set of benchmark alignments, for which a small set of sequences which can be reliably aligned (referred to henceforth as the seed sequences) are provided alongside a much larger set of sequences which are variably distant from the seed (the test sequences). The seed sequences were aligned with (“seed + test alignment”) and without (“seed-only alignment”) the test sequences. We used these benchmark datasets to determine if running the CIAlign cleaning functions can bring the alignment of the seed sequences in the seed + test alignment closer to that of the seed sequences in the seed-only alignment.

A median of 2.10% of correctly aligned residue pairs and 8.22% of residues were removed from the seed sequences in the seed + test alignments, while 92.1% of gaps introduced into the seed sequences were removed (Fig. 7A, Table 2). Regression analysis showed an average loss of 0.130% of correctly aligned residue pairs for every 1% of the alignment removed with CIAlign (Fig. 7B). There was no significant change in seed sequence phylogeny from the seed + test alignment before and after running CIAlign (nRF, p = 0.928, QD, p = 0.672, Mann–Whitney U test) (Fig. 7C, Table 2). Comparing the consensus for the seed sequences in the seed-only alignment with the consensus for the same sequences in the seed + reference alignment, the mean percentage identity increased dramatically by 28.8% after running CIAlign (p = 2.35E−17, Mann–Whitney U test) (Fig. 7D, Table 2).

Figure 7 Metrics from benchmarking CIAlign using HomFam and QuanTest2.

(A) Box plot showing the impact of running CIAlign with moderate settings (Table S2) on the seed sequences in combined alignments of seed and test sequences from the HomFam benchmark set (Wright, 2015), from left to right, the y-axis represents proportion of correctly aligned pairs of residues (Sievers et al., 2013) removed (identified by comparison with alignments of the seed sequences only), proportion of total nucleotides (i.e., non-gap positions) removed, proportion of gaps removed, proportion of positions (gap or non-gap) removed. (B) Scatter plot showing a linear regression analysis of the impact of the total proportion of positions removed on the proportion of correctly aligned pairs of residues removed by CIAlign (identified by comparison with alignments of the seed sequences only) for the HomFam benchmark set. The statistic m is the slope of the regression line (C) Violin plot showing the distribution of normalised Robinson-Foulds distances (Hall, 2008) (nRF) and Quartet divergence (qD) (Fletcher & Yang, 2009) between maximum likelihood trees generated based on seed sequences in alignments of seed sequences only and alignments of seed sequences plus test sequences from the HomFam benchmark set  (Wright, 2015), with (blue) and without (orange) cleaning with CIAlign. (D) Density plot showing the distribution of the percentage identity (top), Needleman-Wunsch score (middle)  (Needleman & Wunsch, 1970) and alignment width between consensus sequences generated from seed sequence only alignments and consensus sequences generated from combined seed and test sequences in the HomFam benchmark set (Wright, 2015). (E) Density plot showing the distribution of the percentage similarity between reference secondary structures and secondary structures based on alignments before (orange) and after (blue) running CIAlign with moderate stringency settings (Table S2), calculated using QuanTest2 (Finn et al., 2014) and using the QuanTest2 reference structures and test alignments. (F) Scatter plot showing a linear regression analysis of the impact of the percentage of the original sequence length remaining after running CIAlign, with moderate parameter values (Table S2), on the change in the percentage of correct positions in the structure prediction after running CIAlign. The statistic m is the slope of the regression line.

Table 2 Table showing the impact of running CIAlign with moderate settings (Table S2) on the seed sequences in combined alignments of seed and test sequences from the HomFam benchmark set (Wright, 2015).

The median proportion of correctly aligned pairs of residues (Sievers et al., 2013) removed (identified by comparison with alignments of the seed sequences only), proportion of total nucleotides (i.e., non-gap positions) removed, proportion of gaps removed, proportion of positions (gap or non-gap) removed were calculated for all HomFam datasets. Normalised Robinson-Foulds distances and Quartet divergences are between maximum likelihood trees generated based on seed sequences in alignments of seed sequences only and alignments of seed sequences plus test sequences from the HomFam benchmark set (Wright, 2015), before and after running CIAlign. Consensus percentage identity is between consensus sequences generated from seed sequence only alignments and consensus sequences generated from combined seed and test sequences in the HomFam benchmark set (Wright, 2015). QuanTest2 percentage similarity is the percentage similarity between reference secondary structures and secondary structures based on alignments before and after running CIAlign with moderate stringency settings (Table S2), calculated using QuanTest2 (Finn et al., 2014) and using the QuanTest2 reference structures and test alignments. All statistics are two-sided Mann Whitney U tests comparing alignments before and after running CIAlign.

Metric	Statistic	Before/After CIAlign cleaning	
		Before	After	
Correct Pairs Removed	Median %	–	2.1	
Nucleotides Removed	Median %	–	8.22	
Gaps Removed	Median %	–	92.13	
Positions Removed	Median %	–	70.38	
Normalised RF Distance	Mean	0.19	0.19	
MWU Test Statistic	–	3542	
MWU P-value	–	0.93	
MWU Significance	–	–	
Quartet Divergence	Mean	0.11	0.11	
MWU Test Statistic	–	3693	
MWU P-value	–	0.67	
MWU Significance	–	–	
Consensus Percentage Identity	Mean	19.77	48.58	
MWU Test Statistic	–	6264	
MWU P-value	–	2.35E -17	
MWU Significance	–	***	
QuanTest2 Percentage Similarity	Mean	67.86	75.99	
MWU Test Statistic	–	17650	
MWU P-value	–	9.35E -20	
MWU Significance	–	***	
Notes.

Significance is shown as *** if the p-value is less than 0.001, ** if the p-value is less than 0.01, * if the p-value is less than 0.05 and—if the p-value is greater than 0.05.

QuanTest2—protein structure prediction

The tool Quantest2 (Sievers & Higgins, 2020) allows benchmarking of alignment quality in terms of its impact on protein secondary structure prediction. We therefore tested the impact of CIAlign on the percentage similarity between reference secondary structures and those predicted based on an alignment with multiple other sequences. We aligned the sequence sets provided in this benchmark and cleaned the alignments with CIAlign (Table S1). A mean of 76.0% of positions in the secondary structure of the reference sequences in the CIAlign cleaned alignment were consistent with the reference structure, compared to 67.9% of positions in the original alignments, a significant improvement of 8.13% (Fig. 7E, Table 2) (p = 9.35E−20, Mann–Whitney U test). A linear regression demonstrated that any cleaning with CIAlign increases, on average, the percentage of correct positions in the resulting structure but that the benefit decreases linearly with the amount of material removed by CIAlign (Fig. 7F).

Benchmarking data availability

Full output tables for the simulations with EvolvAGene, INDELible, BAliBASE, BadRead, HomFam and QuanTest2 are available in Online Tables 1–4 at http://github.com/KatyBrown/CIAlign/benchmarking/tables and the simulated data and alignments at https://github.com/KatyBrown/benchmarking_data_CIAlign.

Comparing alignment tools

In addition to our primary analyses using MAFFT (Edgar, 2004), MUSCLE (Katoh et al., 2002) and Clustal Omega (Sievers & Higgins, 2018), we measured the performance of CIAlign with a number of other alignment tools, including progressive, iterative, non-heuristic, consistency based, HMM-based, context based and phylogeny aware methods (Supplemental Information 1, Table S3).

CIAlign performed similarly with most alignment tools in terms of not excessively removing correctly aligned residues. The mean proportion of correctly aligned pairs removed was 2.80% across all simulations, tools and stringency levels, with a standard deviation of 5.36% (Fig. S3A, Table S3). There was one particular outlier for this metric, with Clustal Omega (Sievers & Higgins, 2018), a HMM-based method, using stringent settings removes a higher proportion of correctly aligned residues for the EvolvAGene nucleotide simulations (median 24.5%). This is the result of a higher proportion of sequences being removed by the remove divergent function, as the mean percentage identity between pairs of sequences in the Clustal Omega alignments is lower (with a mean of 57.9% identity) than the threshold of 65% identity used to remove divergent sequences under the stringent CIAlign settings (Table S1, Fig. S3B).

Otherwise, the extent to which CIAlign will remove positions from an alignment is primarily related to the number of gaps introduced by the alignment software. Amino acid alignments generated with the tool DECIPHER (Wright, 2015) are outliers because this tool introduces fewer and shorter internal gaps (as opposed to terminal gaps) into these alignments than any other tool (under the default settings), which reduces the number of positions meeting the criteria to be removed with either the crop ends or the remove insertions functions (Fig. S3C, Table S3). Across all tools, there is a positive correlation between the proportion of gaps in the input alignment and the proportion of residues (r = 0.793, p = 1.01E−33, Spearman’s ρ), gaps (r = 0.480, p = 4.99E−10), positions (r = 0.890, p = 1.84E−52) and correctly aligned pairs (r = 0.461, p = 3.00E−9) removed (Fig. S3D).

CIAlign does not significantly change the distance between the true phylogenetic tree and the alignment phylogenetic tree for any of the alignment tools (Table S3). It does however improve the consensus sequence significantly (mean 4.68% improvement) in every case except for the DECIPHER amino acid alignments (Fig. S3E) (Mann–Whitney U test, p < 0.05, exact p-values are available in Table S3).

Additional figures showing a full breakdown of the comparisons between alignment tools are available on the CIAlign GitHub page in the benchmarking/Online_Figures directory. These results are summarised in Fig. S3 and Table S3.

Full results for all alignment tools are available in Online Table 5 at http://github.com/KatyBrown/CIAlign/benchmarking/tables and the simulated data and alignments at https://github.com/KatyBrown/benchmarking_data_CIAlign.

Comparison with Gblocks, trimAl and ZORRO

It is not appropriate to compare CIAlign directly with tools intended specifically to identify and remove poorly aligned columns, as it is intended to be complementary to (and, where appropriate, used alongside) such tools. However, we have calculated the proportion of correctly aligned pairs, gaps and residues removed using the default settings for Gblocks (Talavera & Castresana, 2007), trimAl (Capella-Gutiérrez, Silla-Martínez & Gabaldón, 2009) and ZORRO (Wu, Chatterji & Eisen, 2012) as it may be informative for users familiar with another tool to visualise the relative impact of CIAlign on an alignment. All p-values for this section are available in Table S4.

Across the EvolvAGene and INDELible alignments, CIAlign removed a median of 0.188% of correctly aligned pairs with the most relaxed settings, 0.749% with moderate settings and 3.76% with stringent settings (Fig. S4A, Table S4). To compare, Gblocks removed 22.4%, trimAl 1.42% and ZORRO 0.148% (Fig. S4A, Table S4). CIAlign is therefore significantly less deleterious of correctly aligned material than Gblocks at all three stringency levels ,while trimAl falls between the moderate and stringent CIAlign settings for this measure. ZORRO removes slightly less correctly aligned pairs than CIAlign with relaxed settings (Fig. S4A, Table S4). CIAlign removes significantly less positions (7.41%, 8.10% and 9.96% for relaxed, moderate and stringent settings) overall than Gblocks (38.2%) and trimAl (12.8%) at all stringency settings and a similar proportion to ZORRO (7.64%) when run with moderate settings (Fig. S4A, Table S4). A linear regression, showing the relationship between the total proportion of positions removed with each tool and the proportion of correctly aligned residue pairs removed, shows CIAlign with relaxed settings has a similar trade-off between gain and loss of signal to ZORRO (Fig. S4B). For moderate CIAlign settings trimAl and CIAlign are comparable, except with Clustal Omega alignments, where, as discussed above, CIAlign removes a large proportion of divergent sequences and therefore a greater proportion of correct positions. Highly stringent CIAlign settings are between trimAl and Gblocks for this metric, again with the exception of Clustal Omega alignments (Fig. S4B).

None of these tools significantly increased or decreased the distance between trees generated with the test alignments and the true trees except Gblocks, which significantly increased the distance from the true tree with both divergence measures (Fig. S4C, Table S4). Cleaning with CIAlign generates a consensus sequence with 71.5% identity to the true consensus with all three sets of CIAlign parameters, this is significantly higher than any of the other tools (Table S4).

The exact aligned residue pairs removed by CIAlign and the other tools were also compared, to demonstrate the extent to which CIAlign overlaps with and differs from the other tools (Fig. S4D). As Gblocks removes a very large proportion of the alignment, including all gaps, inevitably a large majority of the positions removed by CIAlign are also removed by Gblocks (Fig. S4D). However, CIAlign precisely targets only positions meeting its criteria, removing much less material than Gblocks overall. Compared with trimAl, the most stringent CIAlign settings remove 30.4% unique material (Fig. S4D). At lower stringency settings the majority of pairs removed by CIAlign are also removed by trimAl, but trimAl again has a much more severe impact on the alignment. With ZORRO, while there is a moderate overlap with CIAlign (33.5%, 48.7% and 58.5% for relaxed, moderate and stringent settings, respectively), there is also a large proportion of material (49.5%, 30.7% and 18.0%) which is uniquely removed by CIAlign (Fig. S4D). When comparing ZORRO, Gblocks and trimAl directly with each other, the overlap is much greater, with ZORRO, the most precise of the three tools, removing primarily a subset of the positions removed by trimAl, which are a subset of those removed by Gblocks (Fig. S4D).

These results demonstrate that CIAlign is performing a different role to these three tools, as the locations targetted by CIAlign are only removed by other tools at the expense of large sections of the alignment which CIAlign would leave intact.

Full results for Gblocks, trimAl and ZORRO compared to CIAlign are are available in Online Table 6 at http://github.com/KatyBrown/CIAlign/benchmarking/tables and the data at https://github.com/KatyBrown/benchmarking_data_CIAlign.

Realignment

As alignment tools take into account all the sequences and columns in the input file, the most scrupulous option will always be to unalign and then realign sequences after running a tool such as CIAlign, rather than using the CIAlign output directly in downstream analysis. To test the extent to which using CIAlign outputs directly without realignment could impact results, we removed gaps from the EvolvAGene alignments cleaned with CIAlign with relaxed, moderate and stringent parameter settings and then reran the original alignment tool on the result. We then calculated the sum-of-pairs score (Bahr et al., 2001) treating the realigned file as the true alignment and the CIAlign output as the test alignment. The mean sum-of-pairs score was 0.984, meaning 98.4% of pairs of nucleotides aligned realigned MSA were also aligned in the CIAlign output (Fig. S5). This suggests that while realigning the MSA cleaned with CIAlign is diligent, the effect is likely to be minimal. The full results of this analysis are available in Online Table 7.

Resource and time requirements

Memory and runtime measurements were conducted by randomly drawing alignments from the HomFam benchmark set (Sievers et al., 2013) and measuring the time and memory used for each of the core CIAlign functions. Further measurements were taken by running the CIAlign core functions on an MSA of constant size with different numbers of gaps. The runtime decreases linearly with an increasing proportion of gaps. The results are shown in Fig. S6.

It should be noted that, besides the size of the MSA and its gap content, the runtime is impacted by which combination of functions is applied. For very long MSAs the size of the final image becomes a limiting factor when creating a sequence logo, as the matplotlib library (Hunter, 2007) has restrictions on the number of pixels in one object. We have provided detailed instructions about this limit in the “Guidelines for using CIAlign” on the CIAlign GitHub.

Examples of using CIAlign with biological data

We also used CIAlign to clean real biological data from several online databases, in order to test and demonstrate its usefulness in automated processing of different types of sequencing data.

Cleaning Pfam alignments

The Pfam database provides manually curated seed alignments for over 17,000 protein families, plus much larger automatically generated full alignments containing sequences identified by database searching (Finn et al., 2014). CIAlign cleaning functions were applied to seed and full alignments for 500 Pfam domains and consensus sequences were generated for both alignments, before and after cleaning. Randomly selected sequences from the full alignment were then compared to each consensus. For the full alignments, the mean identity between the consensus sequence and the alignment sequences increased by 10.7% (p = 0.00, Mann–Whitney U test) after cleaning with CIAlign (Fig. 8A). For the seed alignments identity also increased significantly, by 4.89% (p = 0.00, Mann–Whitney U test) (Fig. 8A). After running CIAlign, the full alignment consensus approaches the level of similarity to the alignment sequences which is seen for seed alignment consensus, despite the full alignment having undergone no manual curation (Fig. 8A). Even for the curated seed alignments, cleaning with CIAlign further increases the similarity between the consensus and the aligned sequences. Full results are listed in Online Table 8.

Removing insertions and deletions from human genes

To demonstrate the ability of CIAlign to remove non-majority indels, we used data for 50 indels across over 150 individuals from the 1000 genomes project (Auton et al., 2015), which has annotated insertions and deletions for individual human genomes. In all cases, CIAlign removed all insertions present in a majority of samples and ignored all insertions present in a minority of samples (Fig. 8B). Full results are listed in Online Table 9.

Removing outliers

CIAlign can also be used to remove clear outliers from an alignment, for example prior to phylogenetic analysis. To illustrate this, we ran the CIAlign cleaning functions on data from the mammalian 10K trees project (Arnold, Matthews & Nunn, 2010). Three single-gene trees were identified with clear outliers, the 12S ribosomal gene from primates and the APOB and RAG1 genes from Carnivora. The issues with these trees are shown in Fig. 8C and Fig. S7. CIAlign successfully removed the outlying group, without removing any other sequences, in all three of these cases.

Discussion

We have demonstrated that CIAlign can successfully mitigate the alignment issues caused by non-majority insertions, poorly aligned sequence ends, highly divergent sequences and short sequences and demonstrated this capability on specific examples, simulated and benchmark datasets and large biological datasets. CIAlign has been shown to significantly improve the accuracy of consensus sequences and secondary structure predictions generated from MSAs (Figs. 6C and 7D) It also minimises the detrimental effect of adding additional poorer quality sequences to both benchmark and real alignments (Fig. 7C and 8A). In most cases, the proportion of correctly aligned material removed by CIAlign is minimal.

Figure 8 Metrics from using CIAlign with biological data.

(A) Left, density plots showing the distribution of percentage identity (top) and normalised Needleman-Wunsch score (Needleman & Wunsch, 1970) (bottom) between samples of sequences from the Pfam (Finn et al., 2014) full alignments and consensus sequences generated based on Pfam seed alignments without (dark blue) and with (light blue) CIAlign cleaning and Pfam full alignments without (pink) and with (orange) CIAlign cleaning. Right, box plots showing the alignment total size (top) and number of gaps (bottom) for these four alignments. (B) Left, bar chart showing the size of insertions from the 1000 Genomes data (Auton et al., 2015) used to test the ability of CIAlign to remove insertions and deletions. Right, bar chart showing the proportion of sequences in which these insertions were present in data from 162 individuals and whether they were (pink) or were not (blue) removed by the CIAlign remove insertions function. (C) Left, phylogenetic tree based on an alignment of sequences from the 10k trees project (Arnold, Matthews & Nunn, 2010) for the 12s ribosomal gene in primates. Colours represent known monophyletic groups of primates. Nodes have been collapsed where multiple sequences from the same group formed a monophyletic clade. Sequences annotated with circles were removed by CIAlign. Top-right, tree based on the same alignment after cleaning with CIAlign, which removed the outlying group. Bottom-right, mini alignments showing the effect of running CIAlign on this alignment.

It is important to note that while CIAlign is helpful in mitigating alignment issues, using an appropriate alignment tool and parameters to generate the original alignment is still essential.

Comparison with other software

While the functionality of CIAlign has some overlaps with other software, for example Gblocks (Talavera & Castresana, 2007), ZORRO (Wu, Chatterji & Eisen, 2012) and trimAl (Capella-Gutiérrez, Silla-Martínez & Gabaldón, 2009), the presented software can be seen as complementary to these, with some different features and applications. Our analyses have shown that CIAlign can precisely remove insertions, divergent sequences and poor quality sequence ends without an excessive impact on the rest of the alignment. CIAlign is much more precise than Gblocks and, except under the most stringent settings, also removes substantially less positions than trimAl. Therefore, although a side effect of using these tools may be to remove the specific features targetted by CIAlign, it would be unnecessarily deleterious for users only wanting to target these features to choose Gblocks or trimAl. CIAlign removes slightly more material than ZORRO, but much of the material removed by both tools is unique, indicating that these tools, while similarly precise, are performing different roles. The impact of CIAlign on the structure of trees generated from the cleaned alignments was shown to insignificant. ZORRO and trimAl also had an insignificant impact, while Gblocks had a significant negative impact on tree accuracy. Compared to non-automated tools, for example Jalview (Waterhouse et al., 2009), CIAlign both saves time and increases reproducibility. The visualisation options provided by CIAlign are not, to our knowledge, available in other tools.

Parameters

Having as many parameters as possible to allow as much user control as possible gives greater flexibility. However, this also means that these parameters should be adjusted, which requires a good understanding of the cleaning functions and the MSA in question. CIAlign offers default parameters selected to be often applicable based on our benchmarking simulations and testing with different types of data. However, parameter choice highly depends on MSA divergence and the downstream application. To choose appropriate values it is recommended to first run CIAlign with all default parameters and then adjust these parameters based on the results. Since the mini alignments show what has been removed by which functions it is straightforward to identify the effect of each function and any changes to the parameters which may be required.

Future work

New features are in progress to be added in the future, such as collapsing very similar sequences, removing divergent columns, and making the colour scheme for the bases or amino acids customisable. CIAlign is currently not parallelised, as the most time limiting function, remove insertions, requires information from the entire alignment. However, a future release will incorporate the ability to process more than one alignment in parallel.

Conclusions

CIAlign is a highly customisable tool which can be used to clean multiple sequence alignments and address several common alignment problems. Due to its multiple user options it can be used for many applications. CIAlign provides clear visual output showing which positions have been removed and for what reason, allowing the user to adjust the parameters accordingly. A number of additional visualisation and interpretation options are provided.

Supplemental Information

Supplemental Information 1 Mini alignments and phylogenetic trees showing the application of CIAlign to phylogenetic data, based on Example 4, a subset of Example 3

(A) Input alignment before application of CIAlign, generated using the command “CIAlign infile example4.fastaplot_input”. (B) Output alignment after application of CIAlign, generated using the command “CIAlign infile example4.fastaall remove_divergent_minperc 0.5”. Subplots were generated using the “draw mini alignment function. (C) Phylogenetic tree generated manually using the literature to show the current best estimate for the phylogenetic relationships between these 91 families of metazoa. Relationships are based on the literature listed in the Supp. References. (D) PhyML (Smith, 2019) phylogenetic tree generated under the GTR model plus default settings on the input alignment before application of CIAlign. (E) PhyML (Smith, 2019) phylogenetic tree generated under the GTR model plus default settings on the cleaned alignment after application of CIAlign. In (C–E) branch colours correspond to the labelled phyla, coloured squares indicate class and bold text indicates order. Common names are shown where available.

Click here for additional data file.

Supplemental Information 2 Metrics from benchmarking CIAlign with simulated data, separated by tool

(A–D) Box plots showing the impact of running CIAlign cleaning functions with relaxed (green, R, left), moderate (blue, M, middle) and stringent (red, S, right) parameter values on alignments generated with MAFFT global  (Katoh et al., 2002), MAFFT local (Katoh et al., 2002), MUSCLE (Edgar, 2004) and Clustal Omega (Sievers & Higgins, 2018) of sequences simulated using either EvolvAGene (Bahr et al., 2001) (left) or INDELible (Sievers & Higgins, 2018) (centre) and on the BAliBASE (Thompson, Plewniak & Poch, 1999) benchmark alignments (right), divided into: (A) proportion of correctly aligned pairs of residues (Sievers et al., 2013) removed (identified by comparison with a benchmark alignment), (B) proportion of total nucleotides (i.e. non-gap positions) removed, (C) proportion of gaps removed (D) proportion of positions (gap or non-gap) removed. (E) Scatter plots for each combination of simulation tool and alignment tool showing a linear regression analysis of the impact of the total proportion of positions removed on the proportion of correctly aligned pairs of residues removed by CIAlign for relaxed (green), moderate (blue) and stringent (red) parameter values. The statistic m is the slope of the regression line.

Click here for additional data file.

Supplemental Information 3 Metrics from benchmarking CIAlign with simulated data with different sequence alignment software

(A) Plots showing the impact of running CIAlign cleaning functions with relaxed (green, left), moderate (blue, middle) and stringent (red, right) parameter values on alignments of nucleotide and amino acid sequences simulated using either EvolvAGene (Bahr et al., 2001) or INDELible (Sievers & Higgins, 2018) and aligned with a large number of different alignment tools. From top to bottom, the y-axis represents proportion of correctly aligned pairs of residues (Sievers et al., 2013) removed (identified by comparison with a benchmark alignment), proportion of total nucleotides (i.e. non-gap positions) removed, proportion of gaps removed, proportion of positions (gap or non-gap) removed. (B) Plot showing the pairwise identity between sequences of nucleotides and amino acids in the alignments generated using different alignment tools and simulated using either EvolvAGene (Bahr et al., 2001) or INDELible (Sievers & Higgins, 2018), calculated as the mean of a similarity matrix generated with the CIAlign make_similarity_matrix_input function. Horizontal lines represent the threshold similarity used to remove divergent sequences under stringent (red), moderate (blue) and relaxed (green) CIAlign parameters for the different alignments (for INDELible nucleotide alignments relaxed and moderate are the same). (C) Plot showing the mean number of internal (non-terminal) gaps (purple) per alignment and the mean length of internal gaps (grey) for the alignments of nucleotides and amino acids generated using different alignment tools simulated using either EvolvAGene (Bahr et al., 2001) or INDELible (Sievers & Higgins, 2018). (D) Scatter plots showing the relationship between the metrics from (A) and the proportion of gaps in the alignment, r is the Spearman’s ρ correlation co-efficient and p the associated p-value. (E) Plots showing the mean percentage identity (top) and change in percentage identity (bottom) between consensus sequences generated from EvolvAGene (Bahr et al., 2001) alignments before (orange) and after cleaning with CIAlign with relaxed (green), moderate (blue) and stringent (red) parameter settings (Table S2) and the input sequence to the simulation.

Click here for additional data file.

Supplemental Information 4 Comparison between CIAlign and the alignment masking tools TrimAl, GBlocks and ZORRO

(A) Box plots showing the impact of running other trimming tools–TrimAl (cyan) (Capella-Gutiérrez, Silla-Martínez & Gabaldón, 2009), Gblocks (grey) (Talavera & Castresana, 2007) and ZORRO (orange) (Wu, Chatterji & Eisen, 2012) –and the CIAlign cleaning functions with relaxed (green), moderate (blue) and stringent (red) parameter values on alignments of sequences simulated using either EvolvAGene (Bahr et al., 2001) or INDELible (Sievers & Higgins, 2018). From left to right, the y-axis represents proportion of correctly aligned pairs of residues (Sievers et al., 2013) removed (identified by comparison with a benchmark alignment), proportion of total nucleotides (i.e. non-gap positions) removed, proportion of gaps removed, proportion of positions (gap or non-gap) removed. (B) Scatter plots showing linear regression analyses of the impact of the total proportion of positions removed after running CIAlign (stringent parameters, red; moderate parameters, blue; relaxed parameters, green), TrimAl (cyan), Gblocks (grey) and ZORRO (orange), on the proportion of correct positions removed by the tool for alignments generated with four alignment tools, from left to right MUSCLE (Edgar, 2004), MAFFT local (Katoh et al., 2002), MAFFT global  (Katoh et al., 2002) and Clustal Omega (Sievers & Higgins, 2018). The statistic m is the slope of the regression line (C) Violin plots showing the distribution of normalised Robinson-Foulds distances (Hall, 2008) (left column) and Quartet divergence (right column) (Fletcher & Yang, 2009) between benchmark trees and test trees after running TrimAl, GBlocks and ZORRO versus CIAlign with the three sets of parameter values, for trees based on simulated sequences generated with EvolvAGene (Bahr et al., 2001) (top row) and INDELible (Sievers & Higgins, 2018) (bottom row). Red and black lines show the median and mean respectively. (D) Venn diagrams showing the overlap between aligned residue pairs (Sievers et al., 2013) removed when running CIAlign with relaxed (left), moderate (centre) and stringent (right) parameter settings (Table S2) and those removed when running TrimAl (row 1), GBlocks (row 2) and ZORRO (row 3) and the overlap between aligned residue pairs removed by each possible combination of TrimAl, GBlocks and ZORRO (row 4).

Click here for additional data file.

Supplemental Information 5 Sum-of-pairs scores comparing CIAlign output to the same alignment after removing gaps and realigning

Box plot showing the distribution of the sum of pairs score  (Sievers et al., 2013) comparing the CIAlign output with relaxed (left, green), moderate (middle, blue) and stringent (right, red) (Table S2) parameter settings and the same alignment after removing gaps and then re-aligning with the same software.

Click here for additional data file.

Supplemental Information 6 Time and memory plots

(A, B) Scatter plots showing the relationship between alignment size, measured as n (number of sequences) * m (number of columns) and the time (A) and memory (B) used by CIAlign to run each of the cleaning and mini alignment functions on the alignment, as the mean of four replicate measurements. 25 input alignments were chosen at random from the HomFam (Wright, 2015) benchmark set. All measurements were take n on one Intel Core i7-6700 core with 4 GB of RAM, running at 3.40 GHz. Memory was measured as resident set size. (C) The relationship between number of gaps in an alignment and the time taken to clean the alignment with CIAlign, measured by generating random MSAs of the same size (n = 2000, m = 3000) but with different proportions of gaps.

Click here for additional data file.

Supplemental Information 7 Additional phylogenetic trees showing an application of the remove divergent function

(A) Left, phylogenetic tree based on an alignment of sequences from the 10k trees project (Wick, 2019) for the APOB gene in Carnivora. Colours represent known monophyletic families of Carnivora. Nodes have been collapsed where multiple sequences from the same family formed a monophyletic clade. Sequences annotated with circles were removed by CIAlign. Top-right, tree based on the same alignment after cleaning with CIAlign, which removed the outlying group. Bottom-right, mini alignments showing the effect of running CIAlign on this alignment. (B) As for (A), but for the RAG1 gene in Carnivora.

Click here for additional data file.

Supplemental Information 8 Parameter values

Relaxed, moderate and stringent parameter settings used for benchmarking with EvolvAGene (Bahr et al., 2001), INDELible (Sievers & Higgins, 2018), BadRead  (Sievers & Higgins, 2020), BAliBASE (Thompson, Plewniak & Poch, 1999), HomFam (Wright, 2015) and QuanTest2 (Finn et al., 2014).

Click here for additional data file.

Supplemental Information 9 BadRead results

Table showing the mean percentage identity between the input sequence to BadRead (Sievers & Higgins, 2020) and consensus sequences generated with and without running CIAlign cleaning functions for alignments of good, medium and poor quality simulated reads. All statistics are two-sided Mann Whitney U tests. Significance is shown as *** if the p-value is less than 0.001, ** if the p-value is less than 0.01, * if the p-value is less than 0.05 and –if the p-value is greater than 0.05. The median percentage correct nucleotides removed is the proportion of positions in the alignment which match the input sequence to BadRead which were removed by CIAlign for each set of simulated reads.

Click here for additional data file.

Supplemental Information 10 Metrics from benchmarking CIAlign with simulated and benchmark data

Results of running CIAlign cleaning functions with relaxed, moderate and stringent parameter values on alignments of nucleotide and amino acid sequences simulated using either EvolvAGene (Bahr et al., 2001) or INDELible (Sievers & Higgins, 2018) and aligned with a large number of different alignment tools. For each stringency level, the median percentage of correctly aligned pairs of residues (Sievers et al., 2013) removed (identified by comparison with a benchmark alignment), proportion of total nucleotides (i.e. non-gap positions) removed, proportion of gaps removed and proportion of positions (gap or non-gap) removed have been calculated for EvolvAGene and INDELible. The percentage of gaps is the percentage of gaps over total positions in the input alignment. Number of internal gaps, length of internal gaps and mean pairwise identity are also calculated for the input alignment only. Mean pairwise identity is the mean value of the output of the CIAlign make_similarity_matrix_input function. The mean normalised Robinson-Foulds distance (Hall, 2008) and Quartet divergence (Fletcher & Yang, 2009) are based on comparison with benchmark trees for EvolvAGene and INDELible. Consensus percentage identity is between the input sequence to EvolvAGene and a consensus sequence based on an alignment of the simulated sequences generated by this tool. All statistics are two-sided Mann Whitney U tests comparing the alignment without running CIAlign to the alignment after running CIAlign with the specified parameters. Significance is shown as *** if the p-value is less than 0.001, ** if the p-value is less than 0.01, * if the p-value is less than 0.05 and –if the p-value is greater than 0.05.

Click here for additional data file.

Supplemental Information 11 Comparison between CIAlign and the alignment masking tools TrimAl, GBlocks and ZORRO

Table showing the impact of running other trimming tools –TrimAl (Capella-Gutiérrez, Silla-Martínez & Gabaldón, 2009), GBlocks (Talavera & Castresana, 2007) and ZORRO (Wu, Chatterji & Eisen, 2012)—and the CIAlign cleaning functions with relaxed, moderate and stringent parameter values on alignments of sequences simulated using either EvolvAGene (Bahr et al., 2001) or INDELible (Sievers & Higgins, 2018). For each stringency level and trimming tool, the median percentage of correctly aligned pairs of residues (Sievers et al., 2013) removed (identified by comparison with a benchmark alignment), proportion of total nucleotides (i.e. non-gap positions) removed, proportion of gaps removed and proportion of positions (gap or non-gap) removed have been calculated for EvolvAGene and INDELible. The mean normalised Robinson-Foulds distance (Hall, 2008) and Quartet divergence (Fletcher & Yang, 2009) are based on comparison with benchmark trees for EvolvAGene and INDELible. Consensus percentage identity is between the input sequence to EvolvAGene and a consensus sequence based on an alignment of the simulated sequences generated by this tool. All statistics are two-sided Mann Whitney U tests comparing the alignment without running CIAlign to the alignment after running CIAlign with the specified parameters. Significance is shown as *** if the p-value is less than 0.001, ** if the p-value is less than 0.01, * if the p-value is less than 0.05 and –if the p-value is greater than 0.05.

Click here for additional data file.

Supplemental Information 12 Materials and methods for benchmarking and larger scale examples

Click here for additional data file.

Additional Information and Declarations

Competing Interests

Author Contributions

Data Availability

The authors declare there are no competing interests.

Charlotte Tumescheit and Katherine Brown conceived and designed the experiments, performed the experiments, analyzed the data, prepared figures and/or tables, authored or reviewed drafts of the paper, and approved the final draft.

Andrew E. Firth conceived and designed the experiments, authored or reviewed drafts of the paper, and approved the final draft.

The following information was supplied regarding data availability:

The code is available at Zenodo: Katy Brown & Charlotte Tumescheit. (2022). KatyBrown/CIAlign: v1.0.15 (v1.0.15). Zenodo. DOI:10.5281/zenodo.6330781.

The benchmarking data is available at GitHub: https://github.com/KatyBrown/benchmarking_data_CIAlign.

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
