# Peer review of "CIAlign: A highly customisable command line tool to clean, interpret and visualise multiple sequence alignments"

_PeerJ, doi:10.7717/peerj.12983_

## Round 0.1 · original submission · Major Revisions

Dear authors,

As you can see, the three reviewers are overall quite positive about your work. However, they are asking for some revisions. In particular, they are asking for a more thorough evaluation of your approach.

I'm asking you to take these comments into account when revising your manuscript.

Best wishes

Burkhard Morgenstern

Reviewer 1 ·

Basic reporting

no comment

Experimental design

no comment

Validity of the findings

no comment

Additional comments

The architects of CIAlign present a tool to improve the quality of
Multiple Sequence Alignments (MSAs) by removing badly aligned elements
and to elucidate the meaning of these cleaned MSAs. CIAlign does not
align unaligned sequences from scratch but takes initial MSAs (for
nucleotide or protein sequences) as input. The manuscript is well
written and contains many good elements. The ability to deal with
fragments, in my opinion, is particularly important. The authors state
the time complexity of their algorithm but do not present any
supporting data. They also give real-world resource requirements for
the down-stream analyses but not for CIAlign itself. Using simulated
and real data, the authors demonstrate the effect of their tool
through its effect on the down-stream analyses (phylogeny
reconstruction). The authors provide unit tests for their software,
which is extremely laudable!

- I believe that the effect of CIAlign, putting it simplistically, is
to remove rows and columns from an MSA and not to re-align residues,
that is, to change the relative alignment of residues and gaps, as
in an 'iterative' aligner, like Muscle. If this is true, then this
should be said (more) explicitly.

- The authors should clearly say in their manuscript if the
application is parallelised/multi-threaded or not.

- The authors should mention in their manuscript resource requirements
(time/memory) for their application and present a (rudimentary)
scalability analysis.

- I would like to see which (initial) alignment program benefits
most/least from the effect of CIAlign. The more different
aligners/methodologies (Consistency (T-Coffee), HMM (Clustal-Omega),
Iteration (Muscle), Partial Order (POA), Secondary Structure
(Decipher), etc), the better. In particular, I would like to know
how CIAlign deals with MSAs from a phylogeny-aware aligner like
Prank (Loytynoja, Phylogeny-aware alignment with PRANK. Methods Mol
Biol, 2014). Prank alignments are usually more 'gappy', as opposed
to, for example, Clustal-Omega alignments.

- The purpose of an MSA (usually) is to be used in a down-stream
analysis, so, demonstrating CIAlign's effect on phylogenetic
analyses is apt. However, I would like to know if CIAlign could also
be evaluated using sum-of-pairs analysis for real (not simulated)
data. I do not believe that Pfam seed alignments or simulated
sequences can be treated (or are intended) as reference
alignments. I would therefore like to see traditional benchmarks
like BAliBASE, Prefab or OXBench being used. I suspect, BAliBASE
(Thompson et al, BAliBASE 3.0: latest developments of the multiple
sequence alignment benchmark. Proteins, 2005) may not work, as
removing sequences or columns will not agree with Balibase's scoring
program. However, HomFam (Sievers et al, Making automated multiple
alignments of very large numbers of protein sequences,
Bioinformatics, 2013) might be amenable, and it would be
particularly useful for complexity analysis.

- Another down-stream benchmark methodology that can be used to
evaluate alignments is to use MSAs to construct a secondary
structure prediction (Sievers et al, QuanTest2: benchmarking multiple
sequence alignments using secondary structure prediction,
Bioinformatics, 2019). This would further validate the effect of
CIAlign.

Reviewer 2 ·

Basic reporting

Overall, I thought it was well written and easy to understand the functionality of the created software tool CIAlign

Experimental design

The individual functions of CIAlign were clearly described, and tested with a range of simulated and real data sets. The GitHub page provides clear install instructions and guidelines for use.

Validity of the findings

no comment

Additional comments

This paper describes a new software tool called CIAlign – which is capable of filtering and cleaning sequence alignments, as well as providing a number of nice functions like making a consensus, sequence logos etc. Overall, I think it is a nice tool, and should be accepted for publication in PeerJ, many of us have similar scripts lying about that do some of these functions, but the authors have bought all of these functions into a single software tool that is easy to use, readily available and well documented.

I have only minor comments, these are more likely to be more for future functionality rather than problems hindering publication.

CIAlign appears to revolve around determining divergence/differences from within the alignment – could this be altered to be based on a single references sequence? perhaps designating a sequence within the alignment as the reference. Then decisions such as removal of insertions, divergence, 5’ and 3’ trimming could be made with respect to that reference.

At first I found Figure 1C a little confusing – why is the insertion only present in seq 5 not removed during the remove insertions functionality? Digging deeper, seems that it is by design and due to the flanking parameter, which the user can change, and in addition that particular insertion would be dealt with if you also ran crop ends. On the GitHub page (and/or on the CIAalign help function) – could you give the range of acceptable parameters? Would setting the insertion_min_flank to 0 remove it? And is 0 an acceptable value there.

3 sequences – a couple of times, CIAlign, and sequence alignments in general, were referred to as involving 3 or more sequences – why not 2 or more? Whilst obviously many of the functions will be redundant (as they probably are for 3 seqs as well), some will still be useful for cases of 2 seqs alignment – does CIAlign work with 2 seq alignments? This is an incredibly minor/trivial point.

I managed to get CIAlgin installed very easily, and working just as easily. But struggled getting some of the graphical functions working on larger alignments – this is not an issue for publication – more a query. Obviously, the world is slightly dominated by SARS-CoV-2 at the moment, so I tried an alignment of 1000 CoV-2 seqs of length 29903, but it struggled/crashed (this could easily be me not adapting the params correctly, so again this is not an issue for publication). I was more wondering if the authors expected the graphical components of CIAlign to work easily for long alignments such as that. In particular, I was interested in creating a sequence logo of a particular region only within the alignment (not of the whole alignment) – is that possible current with CIAlign (to focus the SeqLogo on a region).

Typos:

94 – at the either the – two the’s
Intro – paragraphs 123 and 133 – both start with Finally

Reviewer 3 ·

Basic reporting

See review below.

Experimental design

See review below.

Validity of the findings

See review below.

Additional comments

Users of multiple sequence alignment software often provide input sequences that are difficult to align, incomplete, or non-homologous. In some situations it might be useful to automatically remove poor quality sequences from the alignment. The authors describe a program, CIAlign, they created to mask problematic regions of an alignment. They also provide graphics to show how the alignment looks before and after trimming, highlighting regions that were removed. Although this appears useful, the utility of the program beyond creating more aesthetically pleasing alignments was not clear to me as a user. To better demonstrate the utility of CIAlign, I have the following suggestions:


MAJOR COMMENTS

1) The Introduction provides insufficient background about the utility (or lack thereof) of masking multiple sequence alignments. It has previously been shown that masking rarely improves accuracy in downstream applications, and often by only a small amount. Previous approaches to masking (e.g., PMC3260272) are also insufficiently covered (although they are briefly in the Discussion section). In general, the Introduction only weakly motivates the problem statement and instead focuses on the program. Is the goal of CIAlign to remove artifacts, accelerate downstream analyses, or improve the accuracy of downstream results? I recommend reworking the Introduction to focus on existing programs and their known impact on downstream applications, then it should be possible to clearly state how CIAlign fills an existing niche.

2) The impact of CIAlign is not clear from Figure 6. It looks like there was basically no impact from different degrees of masking on RF or Quartet measures of tree distance (histograms in panel 6B). I cannot see a difference between these histograms. I suspect many users would want the program to improve the accuracy of downstream analyses by masking problematic regions. Figure 7 does not make the case for CIAlign because, of course, removal of highly divergent sequences removes long branches from the resulting tree. While the alignment before/after CIAlign looks substantially different, it does not mean the meaningful information derived from the alignment has changed substantially.

3) CIAlign was not benchmarked against other similar programs. I believe it is necessary to perform a characterization of CIAlign relative to competitors.

MINOR COMMENTS

1) Figure 1 could be made more clear by keeping the same sequence number across panels. For example, it is unclear that sequence #1 is removed from panel A to make panel B, because #1 is still the name of a sequence in panel B.

2) The output of a multiple sequence alignment program is often dependent on which input sequences are included. So masking an alignment might not be as successful as removing problematic sequences and then realigning the remaining sequences. The authors may wish to discuss this or try it as an alternative approach.

---

## Round 0.2 · Minor Revisions

Dear Dr. Brown,

we received two reviewer reports on your revised manuscript "CIAlign ... ". I'm pleased to inform you that the reviewers are now, essentially, happy with your manuscript.

Reviewer #3 still has some minor remarks, and I'm asking you to take these comments into account when re-submitting your manuscript. However, I leave it up to you to decide if you want to follow his/her advice or not.

Thanks for submitting your work to PeerJ

Best wishes

Burkhard Morgenstern

Reviewer 1 ·

Basic reporting

The authors have implemented all necessary changes. I was very impressed by the variety of alignment programs and benchmarks included. Thank you especially for including Prank (however only supplemental) and Decipher.

Experimental design

Adequate

Validity of the findings

Convincing

Additional comments

double 'and' in line 102 ...full columns and and full or partial...

Reviewer 3 ·

Basic reporting

See additional comments.

Experimental design

See additional comments.

Validity of the findings

See additional comments.

Additional comments

As previous reviewer #3, I commend the authors for their thorough revisions to reviewer comments on the initial manuscript. I think CIAlign is now presented as a useful tool addressing the important issue of manual curation of sequence alignments. This is reminiscent of an article:
Morrison DA. Why Would Phylogeneticists Ignore Computerized Sequence Alignment? Systematic Biology. 2009;58(1):150-8.

MAJOR

In this revision, the authors added an analysis of multiple sequence alignments per program using simulated sequences (Fig. S3). I found this somewhat difficult to interpret. Broadly, the indel-aware alignment programs have a higher proportion of residues removed, both correct and total. Most other aligners have a similar proportion of residues removed, except DECIPHER. The authors say this is because DECIPHER adds fewer gaps, but Table S3 shows it as being similar to Clustal Omega and some others.

Isn't the goal to remove the fewest correct pairs per residue remove? I would have preferred to see a graph of correct residues removed versus total residues removed and a similar graph for incorrect residues removed. Plotting "Median % Correct Pairs Removed" (y-axis) versus ""Median % Nucleotides Removed" (x-axis) gives a slope of around 3 under "Moderate" CIAlign stringency. This means that every 3% signal+noise I remove I take away 1% of my true signal, right? This slope seems to be higher for relaxed and lower for stringent, suggesting relaxed is better. It would be nice to see these kinds of analyses added to the manuscript.

In general, the trade-off between true positives gained versus true positives lost using CIAlign seems to still be lacking from the manuscript. For example, QuantTest2 showed more correctly predicted residues (76% versus 68%). However, how many residues were lost to make this improvement? That is, for every benchmark the gain in true signal should be placed into perspective of the loss in total signal. This can also be compared across different CIAlign settings (e.g., insertion only versus ends, stringent versus relaxed, etc.).

MINOR

Please report results to a reasonable number of digits that reflects accuracy of the measurement (e.g., 76% or 76.0% instead of 75.99%). I doubt any of these tests were large enough to offer four significant digits of precision.

If I understood correctly, the nucleotide alignments used here are DNA and not non-coding RNA. If the DNA is coding then far better alignments are usually achieved by aligning the codons or translating-aligning-reverse translating. Some programs can do this automatically. If the alignments are non-coding RNA then many of the aligners will handle them differently. I think it is important to show best practices if the goal is to make better alignments. So it would be good to say somewhere that using the aligner correctly is the first step toward good analyses, and CIAlign is for taking the best possible alignment and cleaning it up.

---

## Round 0.3 · accepted · Accept

Dear Dr. Brown,

Thank you for your submission to PeerJ.

I am writing to inform you that your manuscript - CIAlign - A highly customisable command line tool to clean, interpret and visualise multiple sequence alignments - has been Accepted for publication.

Congratulations!

Best wishes

Burkhard Morgenstern, editor